# The Value of Myocardium and Kidney Histopathological and Immunohistochemical Findings in Accidental Hypothermia-Related Fatalities

**DOI:** 10.3390/medicina58111507

**Published:** 2022-10-23

**Authors:** Andreea Alexandra Hleșcu, Adriana Grigoraș, Gabriela Covatariu, Mihaela Moscalu, Cornelia Amalinei

**Affiliations:** 1Legal Medicine Department, Faculty of Medicine, “Grigore T. Popa” University of Medicine and Pharmacy, 700115 Iasi, Romania; 2Department of Morphofunctional Sciences I, “Grigore T. Popa” University of Medicine and Pharmacy, 700115 Iasi, Romania; 3Department of Histopathology, Institute of Legal Medicine, 700455 Iasi, Romania; 4Faculty of Civil Engineering and Building Services, “Gheorghe Asachi” Technical University, 700050 Iasi, Romania; 5Department of Preventive Medicine and Interdisciplinarity, “Grigore T. Popa” University of Medicine and Pharmacy, 700115 Iasi, Romania

**Keywords:** hypothermia, Armanni-Ebstein lesions, sirtuin 1, ubiquitin, hypoxia

## Abstract

*Background and Objectives:* The post-mortem diagnosis of hypothermia is challenging in forensics. The aim of our study was to detect the kidney and heart histopathological changes that occurred in a group of hypothermia-related fatalities. *Materials and Methods:* The cohort included 107 cases identified in the database of our department between 2007 and 2021, which have been associated with extreme cold stress. Demographic and clinicopathological data were collected from the medico-legal reports. Archived tissue samples were evaluated to identify the histopathological features, in routine haematoxylin-eosin (H&E), Periodic acid-Schiff (PAS), and Masson’s trichrome stainings, while cardiac sirtuin1 (SIRT1) and renal ubiquitin (Ub) immunostaining have been performed. *Results:* The majority of cases exposed to low temperatures were males (76%) from rural regions (68.2%) during the cold season. Paradoxical undressing was documented in 9.3% of cases. The common comorbidities included alcoholism (50.5%), neuropsychiatric diseases (10.3%), diabetes mellitus (3.7%), and lung tuberculosis (4.7%). The microscopic heart exam revealed areas of myocardial degeneration (100%), contraction bands (95.3%), fatty change (13.1%) and focal wavy contractile myocardial cells. Basal vacuolisation of renal tubular epithelial cells (Armanni-Ebstein lesions) (21.5%), focal tubular necrosis (7.5%), tubular renal cysts (7.5%), interstitial haemorrhages (5.6%), diabetic kidney disease (3.7%), background benign nephroangiosclerosis (42.1%), variable thickening of tubules and corpuscles basement membranes, capsular space amorphous material, and intratubular casts were identified in kidney tissue samples. Myocardial cells displayed SIRT1 weak expression, with a loss of immunopositivity correlated with areas with contraction bands, while a variable Ub expression was observed in renal corpuscles capsules, proximal, distal, and collecting renal tubules, Henle’s loops, urothelium, and intratubular casts. *Conclusions:* In the context of the current concept that death associated with hypothermia is still a diagnosis of exclusion, our findings suggest that the microscopic exam provides relevant data that support the diagnosis of hypothermia-related fatalities in appropriate circumstances of death. A deeper insight into the histopathologic findings in hypothermic patients may lead to new therapeutic approaches in these cases.

## 1. Introduction

Despite the progress made during recent decades in forensic pathology, the post-mortem diagnosis of hypothermia is still a challenge. Hypothermia describes a condition in which the mechanism of body temperature regulation is overcome by the influences of stress factors and the core body temperature is below 35 °C (95 °F) [1,2]. Usually, death by hypothermia is established on the circumstances of death and supported by few non-specific autopsy gross findings [2,3]. Among them, frost erythema (reddish-brown discolouration of the skin of larger joints), acute gastric erosions (Wischnewsky spots), bright red lividities, and pancreatic haemorrhages, along with paradoxical undressing and signs of “hide-and-die” syndrome, are the most relevant features [1,4,5,6]. The nonspecific microscopic findings that are suggestive of hypothermia diagnosis include cardiac and neuronal hypoxic changes, pancreatic cell necrosis with leukocytes infiltration, along with the vacuolisation of cardiac myocytes, hepatocytes, and renal tubular epithelial cells [1,4,7]. These lesions show variable occurrence according to the duration of survival and associated comorbidities. Although the basal vacuolisation of renal tubular epithelial cells (Armanni-Ebstein lesion) is considered by some authors to have the “same value of diagnostic sensitivity’’ as that of Wischnewsky spots (blackish-brownish spots due to haematinised haemoglobin resulting from autolysed red blood cells from the circumscribed haemorrhages of the gastric glands in the agonal period) in hypothermia-related fatalities, these may be also caused by underlying comorbidities, such as diabetes mellitus [4,8]. Supplementarily, an increase in blood viscosity that induces the occlusion of small blood vessels, microinfarcts, and tissue degenerative changes have been observed in fatal cases associated with cold stress [2].

The period of time of cold exposure and the timing of treatment play key roles, considering that the early recognition and diagnosis of hypothermia are essential to prevent patients’ tissue lesions and fatality. Since cold stress enhances cardiac repolarisation, the current recommendation applied in hypothermia includes the sustained resuscitation of patients with cardiac arrhythmias, independently of body temperature [9], considering that hypothermic cardiac arrest is reversible [10].

Currently, there are some gaps in our knowledge of the pathophysiology of hypothermia and post-mortem hypothermia diagnosis.

Sirtuin1 (SIRT1), a nicotinamide adenine dinucleotide (NAD)-dependent lysine deacetylase, is expressed in various tissues, acting as an important regulator of endocrine functions, which promotes DNA stability, protects the cells from oxidative stress or from different age-related disorders such as tumours and neurodegenerative diseases [11]. It has been recently demonstrated that SIRT1 plays an important role in cardiomyocytes tolerance to ischemia [12]. Furthermore, SIRT1 may represent a new and promising target in the management of neonatal hypoxic-ischemic encephalopathy therapy using hypothermia [13].

Ubiquitin (Ub), a member of the heat shock protein family, is an 8.5 kDa protein containing 76 amino acids [14]. The eukaryotic cell ubiquitination is associated with different processes, including cell cycle regulation, immune reaction, acute inflammatory response, ischemia, and apoptosis [14]. Ub is responsible for the activation of the ubiquitin-proteasome system that has an essential role in the removal and repair of the denatured proteins, which are produced in cells under hypothermic stress [14]. Ub positivity is a characteristic finding in death due to injuries or hypothermia, being overexpressed in different organs, such as the brain, kidney, and skin [14]. Moreover, Ub is considered a valuable marker of diagnosis in patients with cold exposure, especially since recent findings revealed that hypothermia has a neuroprotective effect against cerebral ischemic-reperfusion by the promotion of voltage-dependent anion channel 3 (VDAC3) ubiquitination, associated with Ub overexpression in the renal tubular epithelium [14,15].

Considering their stimulation under stress in various cell types, such as acid-base disturbances, cardiac arrhythmias, hypoxia, DNA damaging factors, and nutrient deficiencies, important roles may be attributed to SIRT1 [13] and Ub [14]. However, a relatively limited number of studies regarding their evaluation in cold stress conditions have been published [13,14].

In this context, our study has been focused on the investigation of the spectrum of the myocardium and renal parenchyma histopathological features in a large group of hypothermia-related fatalities. Supplementarily, we have investigated SIRT1 and Ub immunoexpression in muscular cardiac cells and renal tubular epithelium as potential markers that can be added to the diagnostic tools in hypothermia-related fatalities. These findings may also add new information regarding the mechanisms involved in cold stress, which can be further exploited in hypothermic patients’ management.

## 2. Materials and Methods

The autopsy reports of our department from the last 15 years (2007–2021) have been reviewed in order to select the hypothermia-related cases. The necropsy examination has been associated with toxicological and biochemical investigations and the collection of cardiac and renal tissue specimens for microscopy. The kidney and heart tissue samples were fixed in 10% formaldehyde, embedded in paraffin, and 4–5 μm thick sections from each specimen were cut for H&E, PAS, and Masson’s trichrome stainings, along with SIRT1 and Ub immunohistochemistry. SIRT1 (monoclonal antibody, human, 1F3/Thermo Fisher Scientific), dilution 1/200, with a nuclear staining pattern and Ub (polyclonal antibody, mouse, 00420/Thermo Fisher Scientific), dilution 1/300, with a cytoplasmic staining pattern have been used according to the producer’s recommended method working steps.

The biochemical analyses focused on the measurement of the ethanol concentration in blood and urine. A 10 mL blood sample was collected from the femoral vein, identified in the Scarpa’s triangle, and a 50 mL urine sample was collected from the urinary bladder, in order to determine the alcoholaemia and alcoholuria, respectively. Additionally, a toxicological exam from blood samples has been performed in order to identify drugs, in suspected cases.

The weather information has been registered according to police investigation and local meteorological station reports at the time of exposure.

### Statistical Analysis

The statistical analysis of specific variables has been performed using MATLAB and Statistics and Machine Learning Toolbox (MathWorks, Natick, MA, USA). Continuous variable types have been reported as means ± standard deviation (SD). The numerical variables had a normal distribution. The comparison between analysed groups has been achieved using Student’s T-test or an ANOVA test for the continuous variables. The series homogeneity has been checked regarding the statistical differences between series variability by the Levene test (testing the homogeneity of variances). The correlations between specific parameters have been tested using a Pearson test by evaluation of the r correlation coefficient [16,17]. The qualitative variables have been presented as absolute frequency (*n*) and relative frequency (%), while the comparison between the study groups was based on a Pearson Chi-square test. The significance level calculated in the utilised tests (*p*-value) has been considered as significant for values of *p* < 0.05.

The research had the approval of the Research Ethics Committee of the “Grigore T. Popa” University of Medicine and Pharmacy, Iasi, no. 17275/20 August 2019, consistent with the Declaration of Helsinki: Ethical Principles for Medical Research Involving Human Subjects.

## 3. Results

### 3.1. Demographical Characteristics and Medico-Legal Findings

A cohort of 107 cases of hypothermia-related deaths has been studied. In total, 26 (24%) were females and 81 (76%) were males. The ages ranged between 24 and 91 years, except for a 10-year-old patient. The mean age was 57.95 ± 1.47 (62.5 ± 13.65 for women and 56.5 ± 15.8 for men).

The predisposing factors for hypothermia-related fatalities have included low socioeconomic status (*n* = 37, 34.6%), severe meteorological conditions (*n* = 16, 14.9%), relatively frequently associated with chronic alcoholism (*n* = 19, 17.8%), acute alcoholism (*n* = 24, 22.4%), detected by blood alcohol level > 1.8 g/L, neuro-psychiatric diseases (*n* = 11, 10.3%), outdoor working conditions (*n* = 3, 2.8%), and traumatic injuries (*n* = 2, 1.9%).

The death area and location in hypothermia-related fatalities have been registered in our study group (Table 1). The death took place either at a domicile or in an unheated or poorly heated shelter with no consistent data obtained for the “indoor” temperature at the time of death, either in hospitals, non-built-up areas, or public spaces.

Hospitalisation has been registered in 41 cases (8 female cases; 33 male cases) in our study group, with a mean period of hospitalisation of 3.97 ± 6.53 days. By the comparison of death areas registered for men and women, a statistically significant difference has been identified (Table 1). Additionally, the comparison regarding the living environment and death area showed statistically significant differences *p* < 0.03). By comparison between men and women, the largest percentage of men were deceased in hospital (22.4%), while the largest percentage of women were deceased at home (10.3%) (Table 1). By comparison between urban and rural areas, 22.4% of victims were deceased at home, followed by 17.8% in hospital, and 16.8% in non-built-up areas in rural areas, while most deaths occurred in hospital (10.3%), in urban areas (Table 1).

With one exception, the deaths occurred during the cold season (October to March) in all cases. According to weather information, the influences of outdoor temperature, wind speed, air humidity, wind chill index, and clothing status in hypothermia-related deaths have been evaluated. The mean outdoor temperature, mean wind speed, and mean air humidity index have been calculated (Table 2).

The mean wind chill index value has been calculated by adapting the calculation formula [18,19] for °C, considering air vs. wind speed, producing the “perceived temperature (Table 2). According to the outdoor temperature, the study group showed a dominant distribution of cases between −10 and +10 °C (86.9%), with a maximum between 0 and +5 °C (37.4%). An exception has been that of a woman immersed in cold water in a well in her house yard, which occurred in June, in which drowning has been excluded as the cause of death. Exposure to extreme temperatures was associated with paradoxical undressing documented in less than 10% of cases (Table 2).

Considering that the wind chill index is currently an important piece of data in hypothermia, the statistical analysis in our study group revealed significant correlations between the wind chill index value and outdoor temperature (*r* = 0.67, *p* < 0.01) and wind speed (*r* = −0.52, *p* < 0.01), without significant correlations between the wind chill index value and humidity (r = 0.056, *p* > 0.05).

Ethanol was detected in femoral vein blood samples in 42% of cases (*n* = 45), the maximum value being 4.65 g/L, and in urine, in 24.3% of cases (*n* = 26), with 3.5 g/L as a maximum value. The toxicological examination has been performed in suspected cases, with benzodiazepines (BZDs) and carbamazepine being identified in three cases (Table 2).

Trauma was mentioned in the post-mortem report as an autopsy finding, being considered as a contributing factor for hypothermia-related conditions in 8.4% of cases (*n* = 9), while no sign of old or recent injury had been noticed in the post-mortem report in 32.7% of cases (*n* = 35).

The most common post-mortem findings included frost erythema (*n* = 46 cases, 43%), gastric haemorrhages (*n* = 48 cases, 44.8%), and pancreatic haemorrhages or pancreatitis (*n* = 20 cases, 18.7%).

### 3.2. Myocardium and Renal Histopathological and Immunohistochemical Findings

The autopsy gross findings in heart examination showed: atherosclerosis, cardiac fibrosis, cardiac hypertrophy, and epicardial petechiae (Table 3). The gross findings have been compared with microscopy findings (*p*-value, Table 3). Hematoxylin and eosin (H&E) sections of the myocardium certified the presence of coronary fibrous cap atheroma, added to variable amounts of ischemic myocardial fibrosis (Figure 1) and cardiomyocyte hypertrophy, revealing also areas of myocardial degeneration in most cases (Table 3). Supplementarily, contraction bands and focal wavy myocardial cells, as characteristic signs of myocardial ischemia, have been noted in most cases (Figure 1). The fatty change of cardiac muscle cells has been observed (Figure 1 and Table 3). The immunohistochemical exam revealed SIRT1 cardiomyocyte positivity, with a frequent loss of SIRT1 positivity in areas with contraction bands, with the heterogeneity of immunohistochemistry patterns, exhibiting cytoplasmic and focal nuclear expression (Figure 2).

A strong statistical significance has been observed between atherosclerosis gross findings, associated with other variable myocardium and epicardium changes and their certification by microscopy (*p* = 0.007).

Furthermore, another interesting feature was that contraction bands, considered as markers of hypoxia due to hypothermia, have been more highly associated with high humidity values (93.5%, *p* = 0.001), without significant correlations with wind speed (*p* = 0.545) or wind chill index (*p* > 0.05).

The analysis of the possible association between heart congestion, contraction bands, and blood alcohol level had no statistical significance (*p* > 0.05). Supplementarily, no statistical association between heart microscopic congestion, contraction bands, cytoplasmic vacuolisations, considered as markers of hypoxia due to hypothermia, and alcoholuria have been observed (*p* > 0.05).

The kidneys gross findings have been: stasis, cicatricial retractions, and cysts (Table 3). No data about kidney morphology could be obtained in cases of hemicorporectomy in the conditions of death in non-built-up areas. Although renal stasis has been more highly associated with death in hospital or at a domicile, by comparison with death in public spaces or in non-built-up areas, no statistical significance has been registered (*p* = 0.79).

The renal tissue microscopy lesions in our study group are illustrated in Table 3, with the most important microscopic features being: congestion, benign nephroangiosclerosis, which has been considered a contributing comorbidity, Armanni-Ebstein lesions, and tubular necrosis (Table 3 and Figure 3). Diabetic kidney disease has been certified by microscopy in only 3.7% of cases. Supplementarily, a variable thickening of the renal tubules and corpuscles basement membranes, evident in PAS staining, associated with amorphous material in the urinary space and collecting tubules intraluminal casts, has been observed in some cases (Figure 3).

Ub membrane positive expression in distal and collecting renal tubules in Henle’s loops, along with the immunopositivity of calyces’ transitional epithelium, have been observed, while weaker immunostaining has been noticed in proximal convoluted tubules. Additionally, the protein-like intratubular casts expressed a variable Ub-positive pattern (Figure 4).

Kidney congestion has been more frequently noticed in deaths at domiciles and hospitals, by comparison with deaths in non-built-up or public areas (57.3% vs. 42.7%) but without statistically significant differences (*p* > 0.05). A strong association between gross stasis and microscopy passive congestion in kidney vessels have been also detected (83.2% vs. 85%, *p* = 0.001) without significant differences according to the death area (*p* = 0.069).

Although kidney congestion has been more highly observed in the temperature range from 3.5 to 8.5 °C (Figure 5), no statistical significance has been identified (*p* = 0.99). The analysis of possible associations between Armanni-Ebstein lesions and the temperature range (Figure 5), wind chill index, air humidity, and wind speed have not identified any statistically significant association (*p* > 0.05).

The analysis of the possible association between chronic alcohol abuse and kidney gross features, added to microscopic congestion, benign nephroangiosclerosis, tubular cysts, interstitial haemorrhages, and tubular necrosis, has not revealed any significant associations (*p* > 0.05). Although microscopic kidney haemorrhages have been frequently detected in cases with neuro-psychiatric diseases, diabetic kidney disease, chronic pyelonephritis, and tuberculosis, no statistically significant associations have been identified (*p* > 0.05).

A strong association between chronic alcoholism and Armanni-Ebstein lesions has been identified (52.2%, *p* = 0.01), while no significant association between blood alcohol level and kidney haemorrhages, tubular necrosis, and Armanni-Ebstein lesions has been detected (*p* > 0.05) (Figure 6). Although kidney congestion has been frequently detected in cases with alcoholuria ranging from 1 to 3 g/L, no statistically significant association has been identified (*p* = 0.683) (Figure 7).

## 4. Discussions

The normal body temperature (36.8 ± 0.9 °C) represents a balance between heat loss and heat gain from the environment, added to that given by the basal metabolic rate [18]. Hypothermia occurs mainly when the body has prolonged exposure to subfreezing temperatures (0 °C), but it may be experienced even at above-freezing temperatures when the wind chill factor is increased [18].

Hypothermia grades are progressive, with whole body pathophysiologic lesions, from hypothermia I (mild) (35–32 °C core temperature), hypothermia II (moderate) (<32–28 °C core temperature), hypothermia III (severe) (<28 °C core temperature), and hypothermia IV (profound) (<24 °C core temperature) functionally converting the victim to a poikilothermic status [2,18,20,21].

Death by hypothermia is usually diagnosed according to the circumstances of death and supported by several non-specific gross findings in the autopsy, which include cold erythema (especially in large joints), Wischnewsky spots, localised frostbites, and pancreatic haemorrhages in forensic medicine [6]. However, these classic features do not provide relevant data for the certification of hypothermia diagnosis in some cases.

The microscopy examination may add important information for the certification of gross findings characteristics for hypothermia, with large variability according to the duration of survival and of comorbidities. Convincing examples are that of haemoglobin resulting from the lysed red blood cells of the superficial vessels [1] with the lack of granulocytes inflammation and hyperaemia, features which appear only in rewarming cases [22] or hypoxia signs, such as microscopic foci of myocardial degeneration [1], the accumulation of lipids in cardiomyocytes [23], contraction bands, and Armanni-Ebstein lesions (vacuolisation in renal tubular epithelial cells cytoplasm) [6].

In both humans and rodents, in experimental hypothermic deaths, microscopy has revealed moderate to severe alterations of the cardiac muscle architecture, with tight adherence between cardiac cells, cardiomyocytes fragmentation and degeneration, and variable vacuoles (vacuolar, colliquative myocytolysis, or fatty changes) in different areas of the myocardium [24,25,26]. Although non-specific contraction bands are constantly noticed in myocardial tissue in deaths due to cardiac infarction and traffic accidents, with or without heart injury, these are more frequently noticed in fatal hypothermia [2,25]. These features are in agreement with the findings in our study group, showing contraction bands and focal wavy myocardial cells, along with areas of degeneration or a loss of myocardial architecture in 95.3% of cases. These alterations were associated with cytoplasm vacuolisations as a result of myocyte hypoxia in 13.1% of cases. Moreover, these changes have partially occurred on a background of pre-existent lesions, such as chronic ischemic cardiomyopathy, registered in 48.6% of investigated cases, and have been associated with cardiomyocytes hypertrophy in 18.7% of cases, as pathological factors which accelerate the onset of myocardial hypoxia induced by thermic stress.

Supplementarily, epicardial haemorrhages have been more highly observed in hospitalised cases or deaths in public spaces (2.8% and 1.9%, respectively) considering the high probability of their association with cardiopulmonary resuscitation (CPR) manoeuvres as a constant finding in medico-legal cases [27], according to our professional experience.

As a particular feature of our study, which has been extended to a large group of cases, myocardial contraction bands have been more highly correlated with the humidity level (93.5%, *p* = 0.001) without significant associations with the wind speed or the wind chill index (*p* > 0.05). Supplementarily, no statistically significant associations between heart microscopic congestion, contraction bands, cytoplasmic vacuolisations, and alcoholuria have been found.

The association between contraction bands, different causes of death, and the immunohistochemical expression of hypothermic cardiac markers, such as SIRT1, represents an interesting area of research. SIRT1 is a nicotinamide adenine dinucleotide (NAD)-dependent histone deacetylase, acting as a transcription factor which protects cells against various stress factors [13], being an important regulator of tissue inflammatory response and cisplatin (CDDP)-induced oxidative stress by NF-kB and nuclear-related factor 2 (Nrf2) genes activation [28]. The SIRT1 protein is expressed in the heart, skeletal muscle, brain, kidney, endothelium, liver, spleen, pancreas, and white adipose tissue, exhibiting an overexpressed pattern in cardiovascular, metabolic, and age-related diseases [11].

Increasing evidence shows that SIRT1 is related to cellular hypoxia and heart protection from endoplasmic reticulum stress-related organ damage [29], ageing, myocardial ischemia, and hypertrophy [13]. Scarce literature data are available regarding SIRT1 cardiac expression in victims of hypothermia, due to limitations of large study groups’ collection. The SIRT1 weak cytoplasmic staining of cardiomyocytes containing contraction bands has been noticed in our study group. SIRT1 expression has been lost in specific areas containing contraction bands, in agreement with the results of Morita et al. [30]. This finding has been considered a characteristic immunostaining pattern in hypothermia-related fatalities in comparison with other death causes such as traffic accidents or pneumonia [30]. This feature may be also considered an important tool in hypothermia-related death in forensics, in opposition to death due to myocardial ischemia or infarction, characterised by a lack of SIRT1 expression in modified myocardial tissue [25], in agreement with our results. Our findings may lead to the hypothesis that the myocardial tissue chronic hypoperfusion may be responsible for the patients’ poor prognoses, despite specific therapy, due to the high incidence of pre-existent myocardial ischemic disease, mainly associated with coronary fibrous cap atheroma. The constant finding of contraction bands, associated with the focal loss of SIRT1 expression, has demonstrated the impossibility of heart-appropriate reperfusion in the examined microscopic sections.

Another feature of our study group was that of the relatively long duration of cold exposure, which has been correlated with the failure of therapy applied in these cases, such as vasoactive or antiarrhythmic drugs, active warming fluid therapy, anaesthetics, muscle relaxants, and orotracheal or supraglottic control airways, additionally to CPR manoeuvres [9].

Extensive research has been also performed to identify the histopathological alterations of kidneys in hypothermic stress [4,31,32]. Non-specific lesions, such as congestion, benign nephroangiosclerosis, chronic pyelonephritis, tubular cysts, crystal deposits, variable tubular and corpuscular basement membranes thickening, more evident in PAS staining, various amounts of amorphous material in the capsular space, and intratubular casts have been also observed in our study. These have been associated with Armanni-Ebstein lesions in 21.5% of cases and with tubular necrosis in 7.5% of cases as possible results of thermal-induced renal hypoxia. Analogous findings with that of myocardial cells have been detected in kidneys, consisting of a vacuolar appearance or fatty degeneration, named Armanni-Ebstein lesions, not only in the renal tubular epithelium but also in hepatocytes [6,31], as we have also observed in our study group.

Although Armanni-Ebstein lesions of renal tubular epithelium are evident in H&E staining, as infranuclear vacuoles in tubular epithelial cells, their value in hypothermia certification is still controversial [6]. In this respect, the study performed by Zhou et al., in an experimental model, has shown that a significant metabolic disruption associated with lipid mobilisation from the tissue stores and their deposition in the cytoplasm of renal tubular cells is not specific for hypothermia, as this may occur in other different circumstances, such as alcoholism, diabetes mellitus, and starvation [4]. Despite the missing knowledge regarding the underlying mechanism of cellular vacuolisation in hypothermia, a constant observation is that this finding is frequently associated with fatal hypothermia [6], as demonstrated in our study. However, the contribution of hypothermia to this phenomenon is difficult to quantify, considering other related conditions in our study group, such as alcoholism (50.5% of cases), diabetes mellitus (3.7% of cases), and malnutrition associated with low socioeconomic status (34.6% of cases), which may have been also involved in the development of these lesions.

The analysis of possible associations between Armanni-Ebstein lesions and temperature range, wind chill index, air humidity, and wind speed in our study has not detected any significant associations (*p* > 0.05). A special mention should be added regarding the autopsy diagnosis of Armanni-Ebstein lesions, which may be extremely difficult in advanced putrefaction and autolysis, which lead to a non-specific cytoplasmic vacuolar aspect and the artifactual separation of the epithelium of proximal tubules from their basement membrane [31], although the death circumstances support hypothermia, findings that have been also noticed in our study group. Thus, Armanni-Ebstein lesions, added to selected immunohistochemical markers, may be useful tools to confirm hypothermia-related deaths, in addition to the circumstances of death and gross findings [33]. In this direction, our immunohistochemical study regarding renal Ub expression may add a valuable contribution.

Ub is a member of the heat shock protein family, which may be stimulated by a variety of factors such as mechanical or ischemic stress [14]. Ub’s main function is to repair and eliminate denatured proteins, followed by their transport to the proteolytic system, being extremely useful in the investigation of the physiopathological mechanism by correlations between the survival period of time and the cause of death [14,34]. Its immunoexpression has been studied in different circumstances of death, such as arson, cerebral lesions, asphyxiation, and hypoxia, with very limited studies in hypothermia [34,35,36]. Literature data have reported a close correlation between the thermal aggression intensity and Ub-proteasome pathway activation, exhibited in renal tissue by an increased Ub expression in distal convoluted and collecting tubules, due to the thermal-induced rhabdomyolysis of cardiac muscle cells [34], findings which have been also observed in our study. Moreover, Henle’s loop Ub-positive expression has been identified in our study group.

Supplementarily, the cells of proximal convoluted tubules have shown a weaker Ub positivity in our study, as compared to collecting tubules, in agreement with previous reports [34]. Our observations have additionally registered a particular pattern of staining, with focal membrane expression, as well as cytoplasmic staining, in calyces and collecting tubules. Supplementarily, a weak immunopositivity has been also noticed in the intratubular casts of collecting and distal convoluted tubules. Moreover, Ub expression in renal intratubular casts proved to be exclusively observed in distal convoluted and collecting tubules, with a lack of expression in the proximal convoluted tubules in previous studies [34,37], analogous to our findings, suggesting that its mechanism is more active in distal nephron components.

Accidental hypothermia is often the result of prolonged exposure to low temperatures, specific to the cold season, or by sudden immersion in cold water, extreme age having a rapid fatal fate, due to the perturbations of the mechanisms of thermoregulation in these population groups [1]. Climate factors associated with low socioeconomic status, sparsely populated regions, and outdoor working conditions stand up as the most important risk factors associated with death by accidental hypothermia, especially in sub-arctic regions [38]. Additionally, in our study, excepting a 10-year-old child, the victims had a median age of 57.95 ± 1.47 years, with a dominant percentage represented by men (76%), 61.7% of them suffering from different comorbidities. These comorbidities have been represented by the following conditions: alcoholism, neuropsychiatric diseases, diabetes mellitus, lung tuberculosis, total/partial gastrectomy), and miscellaneous in our study group.

It is worthwhile mentioning that the fatality due to body exposure to thermic stress is also potentiated by other factors, such as the type and/or presence of clothes during the exposure to reduced temperatures, pre-existent pathologies (e.g., neuro-psychiatric diseases), or the associations between cold exposure with alcohol and/or drugs (e.g., benzodiazepine or carbamazepine) [1]. These observations are also supported by the results of our study, considering that the majority of cases have appeared in the cold season (October to March), which is characterised by high humidity and wind speed, characteristics for temperate regions, except for a case which has been related to sudden immersion in cold water.

The analysis of the possible associations between meteorological conditions and hypothermia revealed that, in dry air with good convection and fast evaporation conditions, the human body may resist for a few hours at minus 54.4 °C [39]. However, 17% of patients’ deaths occurred immediately in the case of cold-water immersion, due to uncontrollable hyperventilation and loss of breathing control, associated with increased blood pressure and cardiac output, along with peripheral vasoconstriction [40,41]. This mechanism has been probably involved in the thanatogenesis associated with cold water immersion, identified in a case from our study group.

The high humidity might have also led to a faster and/or deeper hypothermia in our study group by comparison with the involvement of the wind chill index that is currently incriminated in different reports [18,20]. Furthermore, a significant correlation between the value of the wind chill index and outdoor temperature (*r* = 0.67, *p* < 0.01) and wind speed (*r* = −0.52, *p* < 0.01) have been also found in our study group, supporting the previous hypothermia studies [19,20,42].

It is worthwhile mentioning that paradoxical undressing is sometimes registered in hypothermia cases [22]. This phenomenon is attributed to the failure of the hypothalamus to control the body temperature or the paralysis of the vasomotor centre due to reflex vasoconstriction in severe cold exposure. Paradoxical undressing has been also registered in our study group, in 9.3% of cases (three women and seven men), most probably determined by peripheral vasodilation in deep hypothermia, leading to the perception of an extreme heat sensation, with volitional undressing, followed by coma and death [18]. Another variant of paradoxical undressing registered in the literature is that of “hide-and-die” syndrome, with attempted concealment leading to additional abrasions of the knuckles, knees, feet, and elbows [1,6,18].

The alcohol consumption associated with cold exposure accelerates the fatal fate because alcohol is a central anaesthetic but also a potent peripheral vasodilator, favouring increased heat waste without a counterbalanced thermogenesis increase [2]. This association has been also registered in our study, as 42% of cases had a >0 g/L blood alcohol level, with a median value of 1.67 ± 1.04 g/L, combined with BZD use in three cases.

Drug-induced hypothermia is confirmed by numerous studies, the most incriminated drugs being antipsychotics (e.g., olanzapine, risperidone, clozapine, haloperidol, and BZDs) [43,44,45]. BZDs’ hypothermic effect has been attested by different studies, while they also display beneficial effects, which may be exploited in therapy, such as diazepam-induced neuroprotection after global ischemia [46,47]. BZDs exhibit a positive allosteric modulation of the gamma amino butyric acid (GABA)-A receptor [48]. BZ1 receptor containing α1 isoform exhibits sedative effects, while BZ2 containing α2 isoform exhibits anxiolytic and myorelaxant effects [48]. As a consequence of a GABAergic neurotransmission increase, normal thermoregulation is interrupted, with an enhancement of heat loss, resulting in hypothermia [45,49]. Although detected in only 2.8% of cases, our results are in agreement with literature reports, showing that BZDs contribute to hypothermia-related death, as BZDs’ reduced clearance results in prolonged and increased body heat loss [43,47,49].

The limitations of the study have been related to partial information regarding the circumstances of death, the discrimination between lesions by hypothermia from that of alcohol abuse or drug use, comorbidities, and their variable association, and the different periods of time between the cold exposure and death, with different therapeutic interventions and thermic stress complications. Nonetheless, the technical difficulties in achieving the histopathological and immunohistochemical stainings in tissues which have been variably deteriorated by cold, autolysis, and putrefaction phenomena have been serious limitations of our study.

In summary, although none of the characteristic histopathology findings in hypothermia are specific, a collection of gross, microscopic, and immunohistochemical findings are useful in the diagnosis of hypothermic patients and of fatal hypothermia in forensic pathology.

## 5. Conclusions

Histopathological examination plays an important role in hypothermia diagnosis, added to the circumstances of death, along with patients’ medical history. Our results support that poor socioeconomic status and alcoholism or drug use, such as BZDs or carbamazepines, are correlated to their pharmacodynamics and pharmacokinetics related to cold exposure and are identified as additional risk factors for hypothermia.

The gross findings may orientate the diagnosis towards hypothermia if they are supported by data regarding the circumstances of death. Microscopy reveals cardiomyocyte degenerative changes, such as colliquative myocytolysis or fatty changes and contraction bands, are correlated with hypoxia in hypothermia-related fatalities, a process that should be prevented by specific therapy applied to hypothermic patients.

Histopathological examination identifies variable renal tubular and corpuscular lesions, associated with hypoxia and alterations of lipids and carbohydrates metabolisms due to cold stress.

Comorbidities represent a constant finding in hypothermia-related death and may determine difficulties in discriminating the main factors involved in thanatogenesis. As the occurrence and duration of hospitalisation, therapy, and hypothermia complications are largely variable, these may result in extremely complex diagnoses.

Myocardial weak and heterogeneous SIRT1 expression represents a constant finding in hypothermia. Supplementarily, SIRT1 loss in specific areas of cardiomyocytes showing contraction bands can be associated with its intervention in cold stress. SIRT1 loss may suggest its focal depletion in severe hypoxia determined by prolonged exposure to low ambient temperatures.

Variable Ub immunoexpression in renal corpuscles, distal convoluted and collecting tubules, Henle’s loops, intratubular casts, and calyx urothelium represents a constant finding in hypothermia. Ub expression may be related to hypoxia and myocytolysis, which may occur in hypothermia.

Although hypothermia-related fatalities are still characterised by non-specific findings, the currently available tools of investigation might contribute to a diagnosis algorithm with practical applicability, not only in forensics but also in the therapeutic management of hypothermic patients.

## Figures and Tables

**Figure 1 medicina-58-01507-f001:**
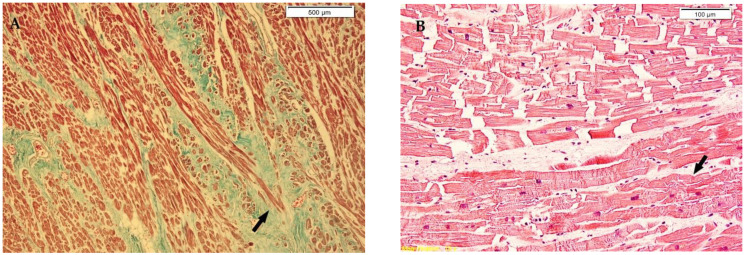
Myocardium microscopic features in hypothermia cases. (**A**) Myocardium with ischemic fibrosis in hypothermia (arrow), Masson’s trichrome staining. (**B**) Loss of normal pattern of organisation in columns of cardiomyocytes, focal wavy myocardial cells (arrow), and degenerative changes with contraction bands, Hematoxylin and eosin (H&E) staining. (**C**) Myocardium with evident wavy myocardial cells (arrow) and few contraction bands, H&E staining. (**D**) Cardiomyocytes vacuolisation (colliquative myocytolysis) (arrow) and blood capillaries, H&E staining ×20.

**Figure 2 medicina-58-01507-f002:**
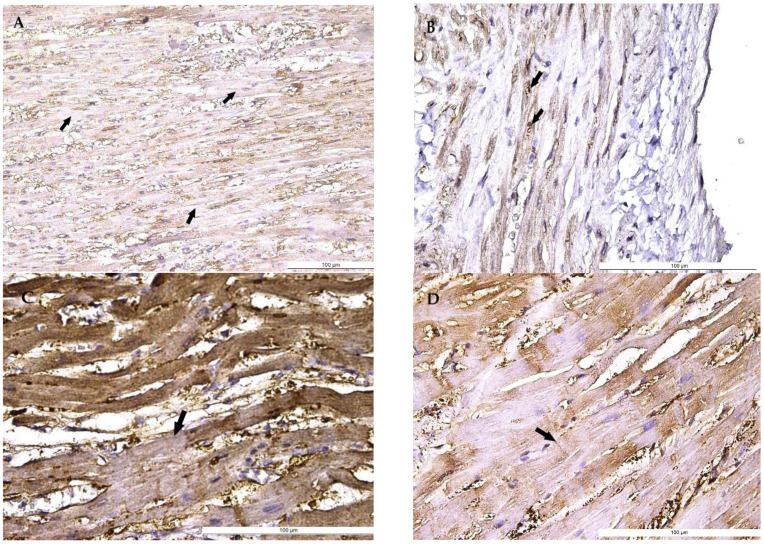
Sirtuin1 (SIRT1) expression in the myocardium of hypothermia cases. (**A**) Nuclear SIRT1 strong positivity along with weak cytoplasmic positivity of cardiomyocytes (arrows). (**B**) Nuclear SIRT1 strong positivity (arrows) along with weak cytoplasmic positivity in myocardial cells. (**C**) SIRT1 positivity in wavy cardiomyocytes, with focal loss of expression (arrow). (**D**) SIRT1 positivity in cardiomyocytes, with focal positivity loss associated with contraction bands (arrow), evident in longitudinal incidences.

**Figure 3 medicina-58-01507-f003:**
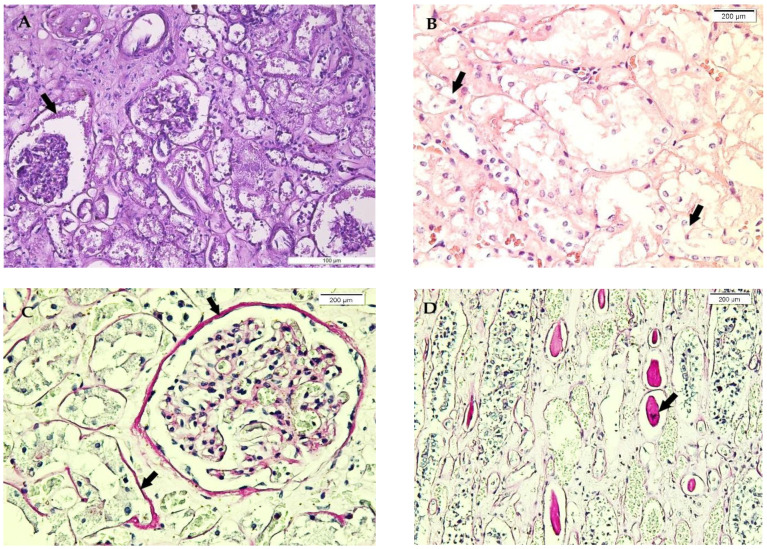
Renal microscopic features in hypothermia cases. (**A**) Focal glomerulosclerosis, interstitial oedema, basement membranes thickening, and amorphous material in Bowman’s space, H&E staining. (**B**) Basal vacuolisation of renal tubular epithelial cells (Armanni-Ebstein lesions) (arrow), H&E staining. (**C**) Renal cortex findings, with corpuscular and tubular basement membranes thickening (arrow), Periodic acid-Schiff (PAS) staining. (**D**) Renal medulla findings, with tubular basement membranes thickening and evident intratubular casts (arrow), PAS staining.

**Figure 4 medicina-58-01507-f004:**
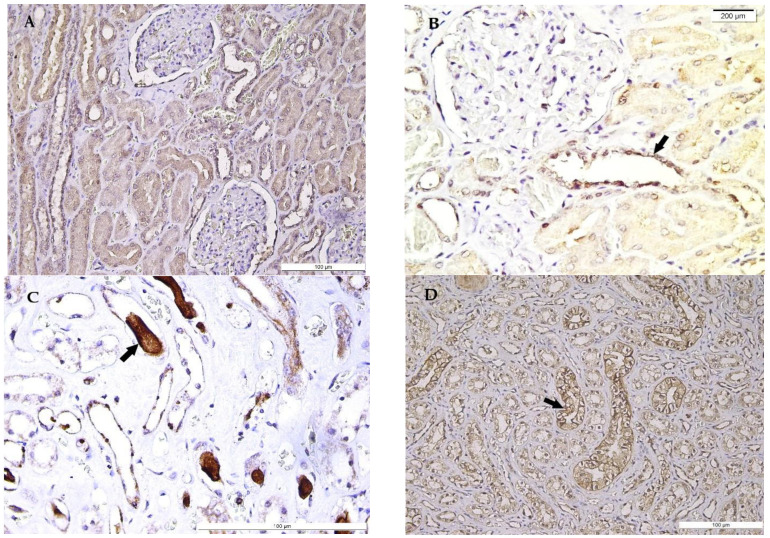
Ubiquitin (Ub) expression in the kidney in hypothermia cases. (**A**) General view of Ub expression in renal corpuscles and proximal and distal convoluted tubules. (**B**) Ub strong expression in renal corpuscle and distal convoluted tubules (arrow), compared with weak expression in proximal convoluted tubules. (**C**) Ub expression in distal convoluted and collecting tubules, Henle’s loops, and intratubular casts (arrow). (**D**) Cytoplasmic and focal membrane Ub expression in renal collecting tubules (arrow) and Henle’s loops epithelium and weak Ub expression in a tubular cast (left top of the image).

**Figure 5 medicina-58-01507-f005:**
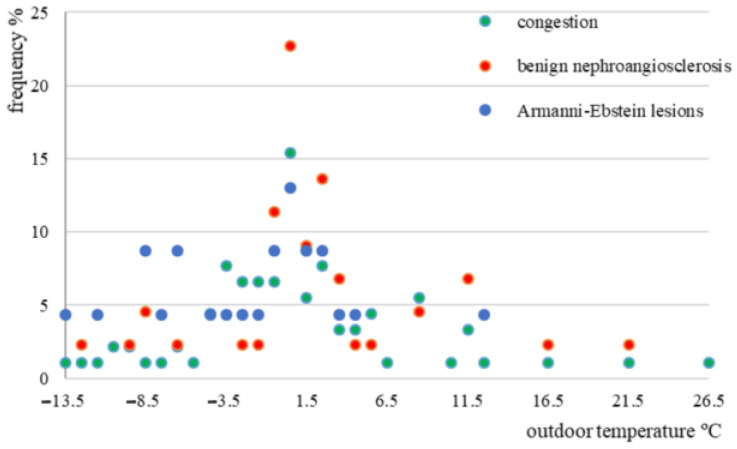
Frequency of cases according to outdoor temperature, congestion, benign nefroangio-sclerosis, and Armanni−Ebstein lesions in the study group.

**Figure 6 medicina-58-01507-f006:**
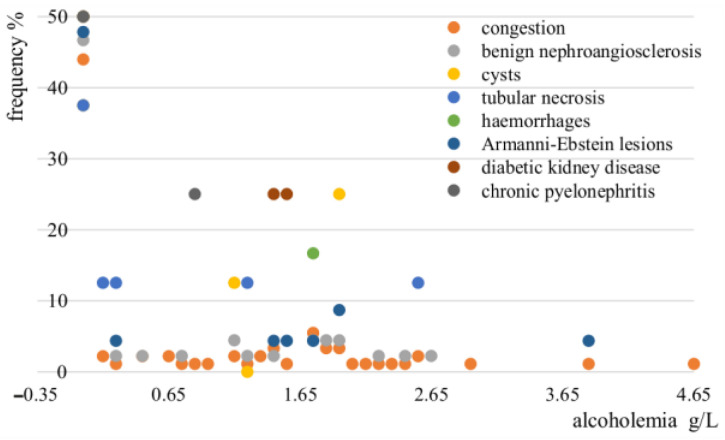
Frequency of cases according to kidney microscopy findings and alcoholemia in the study group.

**Figure 7 medicina-58-01507-f007:**
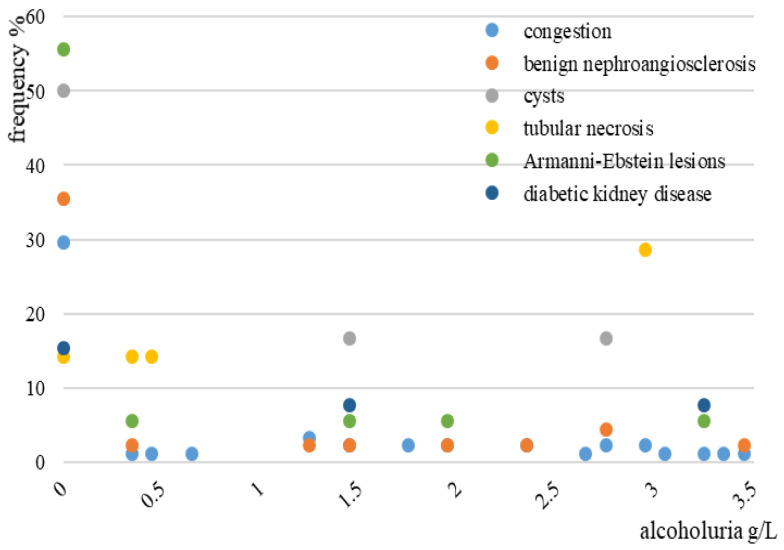
Frequency of cases according to alcoholuria and kidney microscopy findings in the study group.

**Table 1 medicina-58-01507-t001:** The distribution of death area and urban vs. rural location in hypothermia-related fatalities.

Death Area	Total %	Male %	Female %	*p*-Value *	Urban %	Rural %	*p*-Value *
Non-built-up areas	23	19.6	1.9	<0.01	4.7	16.8	<0.01
Hospital	30	22.4	5.6	<0.01	10.3	17.8	<0.01
Domicile	32	19.6	10.3	0.02	7.5	22.4	<0.01
Public spaces	22	14	6.5	<0.01	9.3	11.2	0.03

* Pearson Chi-square test.

**Table 2 medicina-58-01507-t002:** General characteristics of the patients included in our study.

Patients’ Characteristics	No. Cases (%) (*n* = 107)	Mean ± SD	*p*-Value	
Gender				
Female	26 (24%)	-	0.634 ^#^	
Male	81 (76%)	-		
Mean age (years)		57.95 ± 1.47		
Female	26 (24%)	62.5 ± 13.65	0.085 *	
Male	81 (74.8%)	56.5 ± 15.8		
Environment				
Urban	34 (31.8%)	-	0.566 ^#^	
Rural	73 (68.2%)	-		
Mean ambient temperature (°C/°F)	-	0.57 ± 6.4 °C/33 ± 11.5 °F	-	
Mean wind speed (km/h)	-	2.28 ± 3.3	-	
Air humidity index	-	0.74 ± 0.23	-	
Wind chill index **	-	5.83 ± 8.9	-	
Clothing				
Paradoxical undressing Female/Male	10 (9.3%) 3/7 (2.8%/6.5%)	-	0.061 ^#^	
Fully clothed Female/Male	97 (90.6%) 23/74 (21.5%/69.2%)	-	0.015 ^#^	
Comorbidities				
Alcoholism Female/Male	54 (50.5%) 14/40 (13.1%/37.4%)	-	0.021 ^#^	
Neuro-psychiatric diseases Female/Male	11 (10.3%) 4/7 (3.7%/6.5%)	-	0.034 ^#^	
Diabetes mellitus Female/Male	4 (3.7%) 2/2 (1.9%/1.9%)	-	0.892 ^#^	
Total/partial gastrectomy Male	3 (2.8%) 3 (2.8%)	-	-	
Lung tuberculosis Female/Male	5 (4.7%) 2/3(1.9%/2.8%)	-	0.048 ^#^	
Miscellaneous Female/Male	7 (6.5%) 1/6 (0.9%/5.6%)	-	0.037 ^#^	
Without known comorbidities Female/Male	39 (36.4%) 9/30 (8.4%/28%)	-	0.026 ^#^	
Alcoholemia				
Negative Female/Male	62 (57.9%) 18/44 (16.8%/41.1%)	-	0.031 ^#^	
Positive Female/Male	45 (42%) 13/38 (12.1%/35.5%) ^#^	1.67 ± 1.04 g/L 1.71 ± 0.95 g/L/1.52 ± 0.73 g/L *	0.024 ^#^	0.545 *
Alcoholuria				
Negative Female/Male	81 (75.7%) 21/60 (19.6%/56.1%)	-	0.001 ^#^	
Positive Female/Male	26 (24.3%) 8/18 (7.5%/16.8%) ^#^	2.14 ± 1.22 g/L 1.96 ± 1.05 g/L/2.18 ± 0.93 g/L *	0.028 ^#^	0.642 *
Toxicology				
Negative	42 (39.2%)	-		
Positive	3 (2.8%)	-		

# Pearson chi-square test; * ANOVA test; °C–degree Celsius; SD–standard deviation; ** wind chill (°F) = 35.74 + 0.6215 air temperature − 35.75 (wind speed^0.16^) + 0.4275 air temperature (wind speed^0.16^).

**Table 3 medicina-58-01507-t003:** The gross and microscopic features of the heart and kidneys in our study group.

Organ	Gross Findings	No. of Cases *n*, (%); (nF; nM) *	Microscopy	No. of Cases *n*, (%); (nF; nM)	*p*-Value
Heart	Atherosclerosis	65 (60.7%); (17 F; 48 M)	Coronary fibrous cap atheroma	52 (48.6%); (15 F; 37 M)	0.015
	Cardiac fibrosis	40 (37.4%); (8 F; 32 M)	Ischemic myocardial fibrosis	73 (68.2%); (21 F; 52 M)	0.001
	Cardiac hypertrophy	25 (23.6%) (7 F; 18 M)	Cardiomyocytes hypertrophy	20 (18.7%); (3 F; 17 M)	0.020
	Epicardial petechiae	6 (5.6%); (1 F; 5 M)	Epicardial haemorrhages	6 (5.6%); (2 F; 4 M)	-
	No evident lesions	8 (7.5%); (1 F; 7 M)			-
			Degeneration or loss of myocardial architecture	107 (100%); (26 F; 81 M)	-
			Contraction bands	102 (95.3%); (24 F; 78 M)	-
			Congestion	92 (86%); (21 F; 71 M)	-
			Cardiomyocytes vacuolisation (fatty changes)	14 (13.1%); (3 F; 11 M)	-
			Pericarditis	2 (1.9%); (0 F; 2 M)	-
			Myocarditis	1 (0.9%); (1 F; 0 M)	-
			Endocardial degenerative lesions	1 (0.9%); (0 F; 1 M)	-
Kidneys	Stasis	89 (83.2%) (21 F; 68 M)	Congestion	90 (84.1%) (22 F; 69 M)	0.001
	Cicatricial retractions	19 (17.8%) (7 F; 12 M)	Benign nephroangiosclerosis	45 (42.1%) (15 F; 30 M)	0.001
	Absent (hemicorporectomy)	1 (0.9%) (0 F; 1 M)			-
	Renal cysts	9 (8.4%) (2 F; 7 M)	Tubular cysts	8 (7.5%) (1 F; 7 M)	0.003
			Armanni-Ebstein lesions	23 (21.5%) (5 F; 18 M)	-
			Tubular necrosis	8 (7.5%) (2 F; 6 M)	-
			Interstitial haemorrhages	6 (5.6%) (2 F; 4 M)	-
			Diabetic kidney disease	4 (3.7%) (2 F; 2 M)	-
			Chronic pyelonephritis	4 (3.7%) (2 F; 2 M)	-
			Renal crystal deposits	1 (0.9%) (1 M)	-

* F—female; M—male; Pearson Chi-square test.

## Data Availability

All supporting data are provided in the current manuscript.

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
