# Peer review of "The Value of Myocardium and Kidney Histopathological and Immunohistochemical Findings in Accidental Hypothermia-Related Fatalities"

_medicina, 2022, doi:10.3390/medicina58111507_

Round 1

Reviewer 1 Report

Dear Authors,

Thank you for the opportunity to review the manuscript entitled "The value of histopathological and immunohistochemical findings of myocardium and kidney in mortal cases associated with accidental hypothermia." The study attempts to evaluate the pathologies present in the post-mortem examination and their relationship to hypothermia. I value the tremendous amount of work you have put into your study, but the manuscript needs major linguistic correction and rewriting. Also, I suggest consulting the statistics (and results) with a professional statistician. To improve the quality of the manuscript, you should focus on many aspects, including the following: 

1)How may the presented results influence the management of a patient in hypothermia? I suggest expanding on this topic (considering the aims/scopes of Medicina). Otherwise, I recommend submitting to a journal related to forensics.

2) Do the findings show a variable incidence depending on survival time and concomitant diseases? See line 467-469, which I think is very important and has not been adequately highlighted.

Other issues:

Line 77-82: hard to read sentence. Please rephrase. 

Line 77: “Considering their stimulation under stress in various cell types” – what kind of stress? I presume that hypothermia may not be the only cause. Can acid-base disturbances, cardiac arrhythmias, hypoxia, etc., possibly affect SIRT1 and Ub? The question we have to answer in clinical practice during hypothermic cardiac arrest is what came first - CA or hypothermia. This has enormous implications for continuing CPR and prognosis.

Line 103 (Table1):  I do not think it is necessary to present this as a table.

Line 119: the commission's approval number must be provided.

Line 124-130 : The text is extremely unreadable. Massive amount of details regarding age, including fig 1. Consider rewriting the text and removing fig 1

Line 136: “adding up together to 32.70%” useless information (can be calculated) 

Line 137: How do you know it was “acute”? 

Line 138: What’s “working conditions” ?

Line 139: What is “The location of exitus”? The place of death?

143: please remove the phrase “according to Fisher-Freeman-Halton Test (Fisher's test - Conventional Monte Carlo Method)”. 

144: The word “group” is unnecessary - cohort is a group

Line 144-150: I completely fail to understand the content presented in this paragraph. “In our cohort group, hospitalization has been registered in 41 cases” = 38%, but than you write: “22.43% of victims have deceased at home, followed by 17.76% in hospital” ?

Line 156-157: should be in methodology section.

Line 161-165: again, hard to read. Please divide this sentence into shorter parts.

172-173: wind chill is a derivative of temperature and wind speed, why did you check correlations with statistical tests? Or maybe I didn't understand something.

Line 186: Table 3 fully describes the characteristics of the population. I suggest removing all parts of the manuscript body that duplicates this information (line 123-196).

Line 200-202: I don't understand, atherosclerosis “with/without” other pathologies in all 3 most common findings? shouldn't it be "atherosclerosis (%, X/Y), myocardial fibrosis (%, X/Y)" and so on?

Line 217: How is it possible that 65 cases showed "atherosclerosis and myocardial fibrosis" (gross findings) but only 52 on microscopic examination?  

Line 221-223: A very important sentence! Could the other findings also have resulted from stressors other than hypothermia?

Line 226: Why is it interesting that contraction bands were correlated with humidity but not wind speed / wind chill index?

230: What do the % mean, since not the correlation coefficients? Same in line 234.

235: Fig 4A: How can atherosclerosis/hyperthrophy/fibrosis correlate with humidity? Same fig 4B. 

Line 267: “Armanni-Ebstein lesions have been correlated with death at domicile” - given the lack of statistical significance, I suggest not writing such things.

Line 272: again, no statistical significance, so why is it interesting? 

Line 273: what is “temperature frame”?

Line 278: "Furthermore, the statistical analyse revealed a correlation between(...)" - You did not investigate the correlation in statistical tests. Additionally, statistical significance was not found. I suggest to consult the methodology (and results) with a professional statistician.

Ilne 299: The discussion should be rewritten. The beginning contains a lot of unnecessary information - this is not a review. I suggest focusing on the main topic of the study and comparing the results with the available literature.

Line 317-321: I suggest to recheck hypothermia stages with actual literature (!)

Line 342: there is no such thing as “accidental pathological hypothermia” – just "accidental hypothermia". 

Line 494: “As an original finding in our study, statistical analysis showed that Armanni-Ebstein  lesions have been more likely seen in the temperature frame between -9.5-2.5C (16.82%, 495 p=0.95)” – questionable findings, as p=0.95. 

Ine 543-546: I do not believe that these are conclusions that can be drawn from the study.

Line 547-550: this was not the purpose of the study

551-553, 560-562: this are not conclusions form your study

Author Response

Response to Reviewer 1 Comments

We appreciate the positive general comments and the corrections required by Reviewer 1 to the manuscript and we have done all the necessary changes.

General comment:

Thank you for the opportunity to review the manuscript entitled "The value of histopathological and immunohistochemical findings of myocardium and kidney in mortal cases associated with accidental hypothermia." The study attempts to evaluate the pathologies present in the post-mortem examination and their relationship to hypothermia. I value the tremendous amount of work you have put into your study, but the manuscript needs major linguistic correction and rewriting. Also, I suggest consulting the statistics (and results) with a professional statistician. To improve the quality of the manuscript, you should focus on many aspects, including the following: 

  • How may the presented results influence the management of a patient in hypothermia? I suggest expanding on this topic (considering the aims/scopes of Medicina). Otherwise, I recommend submitting to a journal related to forensics.

Response. We agreed with the reviewer and we have made the revision according to the comments (highlighted in red in the Introduction section of the manuscript). Thank you for pointing this out.

  • Do the findings show a variable incidence depending on survival time and concomitant diseases? See line 467-469, which I think is very important and has not been adequately highlighted.

Response. We have added additional information regarding SIRT1 value in hypothermia cases diagnostic (highlighted in red in the Discussion section of the manuscript). Thank you for this suggestion.

Other issues:

  • Line 77-82: hard to read sentence. Please rephrase. 
  • Line 77: “Considering their stimulation under stress in various cell types” – what kind of stress? I presume that hypothermia may not be the only cause. Can acid-base disturbances, cardiac arrhythmias, hypoxia, etc., possibly affect SIRT1 and Ub? The question we have to answer in clinical practice during hypothermic cardiac arrest is what came first - CA or hypothermia. This has enormous implications for continuing CPR and prognosis.

Response. We have revised the text according to the comments (highlighted in red in the Introduction section of the manuscript). Thank you for this suggestion.

  • Line 103 (Table1): I do not think it is necessary to present this as a table.

Response. We agreed with the reviewer and specific changes were made in the Materials and Methods section and Table 1 was removed (highlighted in red in the text of the manuscript). Thank you for this suggestion.

  • Line 119: the commission's approval number must be provided.

Response. We have made the revision according to the comment (highlighted in red in the end of the Materials and Methods section of the manuscript). Thank you for pointing this out.

  • Line 124-130: The text is extremely unreadable. Massive amount of details regarding age, including fig 1. Consider rewriting the text and removing fig 1

Response. We agreed with the reviewer and specific changes were made in the 3.1 section and the Figure 1 was removed (highlighted in red in the text of the manuscript). Thank you for this suggestion.

  • Line 136: “adding up together to 32.70%” useless information (can be calculated)
  • Line 137: How do you know it was “acute”? 
  • Line 138: What’s “working conditions” ?

Response. We agreed with the reviewer and specific changes were made according to the comments (highlighted in red in the 3.1. section of the manuscript). Thank you again for these suggestions.

  • Line 139: What is “The location of exitus”? The place of death?

Response. We have replaced the word in the main text of the manuscript (highlighted in red in the 3.1. section). Thank you for pointing this out.

  • Line 143: please remove the phrase “according to Fisher-Freeman-Halton Test (Fisher's test - Conventional Monte Carlo Method)”.

Response. We agreed with the reviewer and replaced it with the correct name of the test.

  • Line 144: The word “group” is unnecessary - cohort is a group

Response. We agreed with the reviewer and have used only a term. Thank you for this suggestion.

  • Line 144-150: I completely fail to understand the content presented in this paragraph. “In our cohort group, hospitalization has been registered in 41 cases” = 38%, but than you write: “22.43% of victims have deceased at home, followed by 17.76% in hospital” ?

Response. We agreed with the reviewer and have made the revision of the text according to the comments (highlighted in red in the 3.1. section of the manuscript). As special mention, the main text of the manuscript has been revised by a first-language English speaker. We appreciate these suggestions.

  • Line 156-157: should be in methodology section.

Response. Thank you again for this comment. The text has been moved to the methodology section (highlighted in red in the manuscript).

  • Line 161-165: again, hard to read. Please divide this sentence into shorter parts.

Response. We agreed with the reviewer and specific changes were made in 3.1. section (highlighted in red in the text of the manuscript).

  • 172-173: wind chill is a derivative of temperature and wind speed, why did you check correlations with statistical tests? Or maybe I didn't understand something.

Response: Considering that, the wind chill index is currently an important data in hypothermia, according to literature data (mentioned in the text and in the Reference section of the manuscript), we decided to check its statistical correlations in our study group.

  • Line 186: Table 3 fully describes the characteristics of the population. I suggest removing all parts of the manuscript body that duplicates this information (line 123-196).

Response. Thank you for pointing this out. We agree with the reviewer and reorganized the text of the 3.1 section of the manuscript by removing the duplicated information according to table 3 (now Table 2).

  • Line 200-202: I don't understand, atherosclerosis “with/without” other pathologies in all 3 most common findings? shouldn't it be "atherosclerosis (%, X/Y), myocardial fibrosis (%, X/Y)" and so on?
  • Line 217: How is it possible that 65 cases showed "atherosclerosis and myocardial fibrosis" (gross findings) but only 52 on microscopic examination?  

Response. We agree with the reviewer and reorganized table 2 and the text of the manuscript in order to present the results according to the suggestions (Please see section 3.1. and table 3). We appreciate these suggestions.

To increase the precision of the data presented, the gross and microscopic features have been revised (Please see table 3 and the 3.2 section of the manuscript). Considering the evolution of atheroma, only the stage of fibrous cap atheroma has been recorded in the microscopy reports, resulting in a different number of cases between general gross findings and histopathological reports.

Line 221-223: A very important sentence! Could the other findings also have resulted from stressors other than hypothermia?

Response. Epicardial haemorrhages have been more likely observed in hospitalized cases or public spaces deaths, considering the high probability of their association with cardiopulmonary resuscitation (CPR) manoeuvres, being a constant finding in medico-legal cases, according to our professional experience (highlighted in red in 3.2. section of the manuscript).

Thank you for pointing this out.

- Line 226: Why is it interesting that contraction bands were correlated with humidity but not wind speed / wind chill index?

-  Line 230: What do the % mean, since not the correlation coefficients? Same in line 234.

- Line. 235: Fig 4A: How can atherosclerosis/hyperthrophy/fibrosis correlate with humidity? Same fig 4B.

Response. We agreed with the reviewer and specific changes were made in 3.2. section (highlighted in red in the text of the manuscript). We appreciate this suggestion.

  • Line 267: “Armanni-Ebstein lesions have been correlated with death at domicile” - given the lack of statistical significance, I suggest not writing such things.

Response. We agreed with the reviewer and we removed the text from the manuscript. We appreciate this suggestion.

-  Line 272: again, no statistical significance, so why is it interesting? 

-  Line 273: what is “temperature frame”?

  •  Line 278: "Furthermore, the statistical analyse revealed a correlation between (...)" - You did not investigate the correlation in statistical tests. Additionally, statistical significance was not found. I suggest to consult the methodology (and results) with a professional statistician.
  • Response. We agree with the reviewer and the ''temperature frame'' have been replaced with ''temperature range'' (highlighted in red in 3.2 section of the manuscript). The statistical analysis has been revised by professional Medical statistics. Thank you for pointing this out.
  •  
  • In line 299: The discussion should be rewritten. The beginning contains a lot of unnecessary information - this is not a review. I suggest focusing on the main topic of the study and comparing the results with the available literature.

Response: We have made the revision of the Discussion section according to the comments. The paragraphs have been reformulated, relocated, deleted or shortened. Thank you for pointing this out.

  • Line 317-321: I suggest to recheck hypothermia stages with actual literature (!)

Response. We agreed with the reviewer and added a new reference related to the data presented (Ibrahim et al. 2021, DOI: 10.1080/20961790.2021.1886656).

Line 342: there is no such thing as “accidental pathological hypothermia” – just "accidental hypothermia".

Response. We agreed with the reviewer and removed the word “pathological”. Thank you again for this suggestion.

  • Line 494: “As an original finding in our study, statistical analysis showed that Armanni-Ebstein lesions have been more likely seen in the temperature frame between -9.5-2.5C (16.82%, 495 p=0.95)” – questionable findings, as p=0.95. 

Response. Thank you for pointing this out. The statistical analysis has been revised by a professional specialist in Medical statistics.

- Line 543-546: I do not believe that these are conclusions that can be drawn from the study.

- Line 547-550: this was not the purpose of the study

- 551-553, 560-562: this are not conclusions form your study

Response. We agreed with the reviewer and specific changes were made in the Conclusion section (highlighted in red in the text of the manuscript).

Thank you again for taking the time to review and comment on our manuscript.

Reviewer 2 Report

Please find my comments in the attachment

Author Response

Response to Reviewer 2 Comments

We appreciate the positive general comments and the corrections required by Reviewer 2 to the manuscript and we have done all the necessary changes.

General comment:

Thank you for the pleasure and honor to peer review the extensive manuscript The value of myocardium and kidney histopathological and immunohistochemical findings in accidental hypothermia-related fatalities. As a clinician, and as researcher interested in this field, this pathology study was interesting, while at the same time I am afraid to be unable to comment on the details of the examinations performed. Some general remarks and suggestions may be worthwhile to be considered.

The objective of the study seems to be to confirm the postmortem diagnosis of accidental hypothermia. The study is based on information regarding the environmental circumstances, and the pathological changes of myocardium and kidney, such as gross findings, microscopic findings using several staining techniques, immunohistochemistry, anti-body detection, as well as biochemical and toxicological findings.

Because there is no well-defined research question, it remains vague what the actual objective of the study is. As a result, the manuscript includes a lot of information and data, while it seems to lack a clear conclusion.

At the same time, the authors have made immense efforts to provide an all-inclusive narrative review of the literature on accidental hypothermia that distracts from the objective of the paper.

According to its guidelines for authors, Medicina has no restrictions on the length of manuscripts, provided that the text is concise and comprehensive. This is however not the case. The essence of this study can be addressed in some 30-50% of the current length of the paper without losing any relevant information related to the purpose or aim of the study.

Response: We agreed with the reviewer and we have made the revision according to the comments. The paragraphs have been reformulated, relocated, deleted or shortened in order to maintain the fluent style of the manuscript (highlighted in red in the text of the manuscript). Please see response 2.

Some suggestions to achieve this:

  1. Include a brief summary of the many techniques used (including restrictions in use and value) in the method section as far as relevant for this study (so no narrative review on each technique). This can substantially reduce the information on these techniques in the discussion. This will also keep the focus of the discussion on the data.

Response 1. We agreed with the reviewer, and specific changes were made in the Materials and Methods section and Table 1 was removed (highlighted in red in the text of the manuscript). Thank you for this suggestion.

  1. In the result section, written data and data in tables 1-5, figures and 7 are largely overlapping, redundant; and may be not always relevant for the objective of the study. It may be considered to include some of them as addendum or offer to have these data available on request. It is not sure if all 24 figures in 3, 5 and 6 are relevant for this paper. Most likely, pathologists will know how the microscopic observations look. For non-pathologists, it may be interesting to point an arrow to where to look at.

Response 2. We appreciate the reviewer's comment. The revised manuscript includes 3 tables and 7 figures (16 microscopy figures). Thank you for pointing this out.

  1. Some findings that are described seem to fit in the category nice to know, but are not needed to know for the objective or conclusion of the study.

Response 3. We agreed with the reviewer and specific changes were made in 3.1. and 3.2 section and Table 1 were removed (highlighted in red in the text of the manuscript). Thank you for this suggestion.

  1. A discussion in general starts with the most relevant findings of the study. Most information between lines 298 and 385 are a narrative review that distract from the objective of the study.

It may be considered, as a suggestion, to start the discussion as This study observed the following pathologist findings in the myocardium of persons who died from accidental hypothermia a, b and c. Pathological finding In the kidney showed d, e and f. This is followed by comments on these observations based on literature data.

The next paragraph may focus on the relationship found with the environmental circumstances; again, followed by comments based on previous studies. It needs no, or very limited attention, to irrelevant observations

The third paragraph may refer to confounding factors and comorbidities that may have influenced the findings, followed by comments based on previous studies.

The final paragraph collects all the limitations which are now spread over methods, results and discussion.

Response 4. Thank you for pointing this out. We agree with the reviewer and reorganized the Discussion section. The paragraphs have been reformulated, relocated, or shortened.

  1. It is a common problem when deeply involved in a study with a lot of passion and curiosity, while striving for perfectionism, to end in a manuscript that is overly extensive. It may help to google “Kill your darlings” (related to research, not to poetry) to understand how to reduce the size of the paper without losing content. As a clinician, the value of the paper is the confirmation that hypothermia indeed results in a multi-organ failure once the patient has left the emergency department with a normal temperature. I believe there are a few publications that refer to this and they may be added to the references if this observation is included in the paper.

Response 5. We agree with the reviewer and reorganized the Results section by adding additional information regarding the value of our study results. A specific observation regarding the role of SIRT1 in the hypothermia patient’s diagnosis has been added in the Discussion section (highlighted in red in the text of the manuscript). Thank you for pointing this out.

Some details

  1. Is it a co-incidence that the % of patients with diabetes mellites is the same as microscopic findings of dlabetic kidney disease (3.74%)

Response 1. The microscopic findings associated with diabetic kidney disease have been certified by histopathology in diabetes mellitus patients which have been selected in our study group, as shown in Tables 2 and 3. Thank you for pointing this out.

  1. There is no need to report figures in 2 decimals.

Response 2. We agreed with the reviewer and specific changes were made, where appropriate (Please see Fig. 5 and 7). Thank you again for this suggestion.

  1. It seems that there is only one drowned hypothermic person included. It may be considered to remove this person from the study population. Immersion hypothermia in general occurs much faster than outdoor accidental hypothermia. In addition, there is the other impact of hypoxia. At the same time, it reads as some of the microscopic observations are linked to hypoxia. This is confusing in case there is no drowning component in the circumstances.

Response 3. We expanded the text associated with this particular case in the Discussion section (highlighted in red in the text of the manuscript). We appreciate this suggestion.

  1. Letter-typing, size and line-distance seem to vary occasionally in the manuscript.

Response 4. We have revised and corrected the main text. Thank you for pointing this out.

Thank you again for taking the time to review and comment on our manuscript.

Round 2

Reviewer 1 Report

Dear authors,

Despite some improvements (including statistics), the manuscript still requires major revisions. There are parts of the text that are very difficult to understand and I doubt the quality of the language correction. The conclusions presented do not follow from the study. 

major concerns:

line 75: Currently, there are some gaps in our knowledge of the pathophysiology and diagnosis of hypothermic patients  - I assume you mean "post mortem hypothermia diagnosis"? Clinical diagnosis is based on modified Swiss classification and core temperature measurement.

Line 75 and 83: please use full names (SIRT1 and Ub)

Line 101: Since cold stress enhances cardiac repolarization abnormalities (I suggest to write “affect cardiac repolarization”), the current recommendation applied in hypothermia includes sustained resuscitation of patients with cardiac arrhythmias (“cardiac arrest”), independently of body temperature [14] - Please refer to current ERC resuscitation guidelines.

Line 109: as potential markers that can be added to the diagnosis tools in hypothermic patients – I do not think that histopathological examination is reasonable in the diagnosis of hypothermia in living patients…

Line 184: please check the equation (temp abbreviation). 

Line 191: The wind chill index is calculated from a mathematical equation that includes ambient temperature and wind speed (y=ax+b…). I seriously doubt whether it is rational to show a correlation between these variables - it is an equation.

Line 240: considering the high probability of their association with cardiopulmonary resuscitation (CPR) manoeuvres, as a constant finding in medico-legal cases, according to our professional experience. - these are not the results of this study, rather speculation, should be in the discussion.

Line 249: This finding may suggest that high humidity might had led to a faster and/or a deeper hypothermia in our study group by comparison with wind chill index that is currently incriminated in different reports [17,18]. – again, this statement should be in the discussion.

Line 322-325: I suggest using the clinical classification of accidental hypothermia  35-32, <32-28, <28 - please check publications by Ken Zafren or Peter Paal (10.3390/ijerph19010501)

Line 339: In both human and experimental hypothermic death – what is experimental hypothermic death? Animal studies?

Line 374: Our observations have noticed SIRT1 cytoplasmic staining of cardiomyocytes containing contraction bands but with its loss in specific areas with contraction bands, in agreement with the results of Morita et al., as compared with other causes of death, such as traffic accidents or pneumonia – hard to read sentence, please rephrase.

Line 388: Another feature of our study group was that of relatively long duration of time of cold exposure, which has been correlated with failure of specific therapy. - What kind of therapy? Do you know the details of the CPR?

Line 463: However, if the humidity is 100% or if the body is immersed in water, the body temperature starts to increase  when the ambient temperature is over 34.4oC  - increase? Over? 

Line 515: The conclusions described in the paragraph are NOT based on the study and are speculation. Most of this text should be in the discussion. Please stick to the objectives of the work - histopathological changes and not side issues (weather, exposure etc). The conclusions must be completely rewritten and shortened.

Line 517: Our results support that environment cold weather associated with high humidity are the main predisposing factors which lead to failure of specific therapy and to hypothermia-related fatalities. – again, what specific therapy?

Line 536: Myocardial weak and heterogeneous SIRT1 expression, along with its loss in specific areas of cardiomyocytes showing contraction bands represents a constant characteristic in hypothermia in correlation with its intervention in cold stress, suggesting its focal depletion in severe hypoxia determined by prolonged exposure to low ambient temperatures. – hard to read, I do not fully understand the meaning of this sentence. (same 541-544, try to make it more readable)

Author Response

Response to Reviewer 1 Comments

We appreciate the positive general comments and the corrections required by Reviewer 1 to the manuscript and we have done all the necessary changes. Thank you again for your valuable comments.

Dear authors,

Despite some improvements (including statistics), the manuscript still requires major revisions. There are parts of the text that are very difficult to understand and I doubt the quality of the language correction. The conclusions presented do not follow from the study. 

major concerns:

  • line 75: Currently, there are some gaps in our knowledge of the pathophysiology and diagnosis of hypothermic patients  - I assume you mean "post mortem hypothermia diagnosis"? Clinical diagnosis is based on modified Swiss classification and core temperature measurement.

Response: We agreed with the reviewer and we have made the revision according to the comments (highlighted in red in the Introduction section of the manuscript). Thank you for pointing this out.

  • Line 75 and 83: please use full names (SIRT1 and Ub)

Response: We have revised the text according to the comments (highlighted in red in the Introduction section of the manuscript). Thank you for this suggestion.

  • Line 101: Since cold stress enhances cardiac repolarization abnormalities (I suggest to write “affect cardiac repolarization”), the current recommendation applied in hypothermia includes sustained resuscitation of patients with cardiac arrhythmias (“cardiac arrest”), independently of body temperature [14] - Please refer to current ERC resuscitation guidelines.

Response: We agreed with the reviewer and specific changes were made in the Introduction section (highlighted in red in the manuscript). Supplementary, a new reference related to the current ERC resuscitation guidelines has been added.

  • Line 109: as potential markers that can be added to the diagnosis tools in hypothermic patients – I do not think that histopathological examination is reasonable in the diagnosis of hypothermia in living patients…

Response: We have revised the text according to the comments (highlighted in red in the Introduction section of the manuscript). Thank you for pointing this out.

  • Line 184: please check the equation (temp abbreviation). 

Response: The wind chill index formula has been revised (highlighted in red in the 3.1. section of the manuscript).

  • Line 191: The wind chill index is calculated from a mathematical equation that includes ambient temperature and wind speed (y=ax+b…). I seriously doubt whether it is rational to show a correlation between these variables - it is an equation.

Response: We have revised the text according to the comments (highlighted in red in the 3.1. and Discussion sections of the manuscript). Thank you for pointing this out.

  • Line 240: considering the high probability of their association with cardiopulmonary resuscitation (CPR) manoeuvres, as a constant finding in medico-legal cases, according to our professional experience. - these are not the results of this study, rather speculation, should be in the discussion.

Response: We agreed with the reviewer and the text has been moved to the Discussion section (highlighted in red in the text of the manuscript). Thank you for this suggestion.

  • Line 249: This finding may suggest that high humidity might had led to a faster and/or a deeper hypothermia in our study group by comparison with wind chill index that is currently incriminated in different reports [17,18]. – again, this statement should be in the discussion.

Response: We agreed with the reviewer and the text has been moved to the Discussion section (highlighted in red in the text of the manuscript).

  • Line 322-325: I suggest using the clinical classification of accidental hypothermia  35-32, <32-28, <28 - please check publications by Ken Zafren or Peter Paal (10.3390/ijerph19010501).

Response: We agreed with the reviewer and we have made the revision according to the comments (highlighted in red in the Discussion section of the manuscript). Both indicated references have been added to the data presented. Thank you for pointing this out.

  • Line 339: In both human and experimental hypothermic death – what is experimental hypothermic death? Animal studies?

Response: We have revised the text according to the comments (highlighted in red in the Discussion section of the manuscript). Thank you for pointing this out.

  • Line 374: Our observations have noticed SIRT1 cytoplasmic staining of cardiomyocytes containing contraction bands but with its loss in specific areas with contraction bands, in agreement with the results of Morita et al., as compared with other causes of death, such as traffic accidents or pneumonia – hard to read sentence, please rephrase.

Response: We have revised the text according to the comments (highlighted in red in the Discussion section of the manuscript).  We appreciate these suggestions.

  • Line 388: Another feature of our study group was that of relatively long duration of time of cold exposure, which has been correlated with failure of specific therapy. - What kind of therapy? Do you know the details of the CPR?

Response: We expanded the text with supplementary data related to the specific therapy in hypothermia cases. Thank you for pointing this out.

  • Line 463: However, if the humidity is 100% or if the body is immersed in water, the body temperature starts to increase when the ambient temperature is over 34.4oC  - increase? Over? 

Response: We agreed with the reviewer and we have made the revision according to the comments (highlighted in red in the Discussion section of the manuscript). We appreciate these suggestions.

  • Line 515: The conclusions described in the paragraph are NOT based on the study and are speculation. Most of this text should be in the discussion. Please stick to the objectives of the work - histopathological changes and not side issues (weather, exposure etc). The conclusions must be completely rewritten and shortened.
  • Line 517: Our results support that environment cold weather associated with high humidity are the main predisposing factors which lead to failure of specific therapy and to hypothermia-related fatalities. – again, what specific therapy?
  • Line 536: Myocardial weak and heterogeneous SIRT1 expression, along with its loss in specific areas of cardiomyocytes showing contraction bands represents a constant characteristic in hypothermia in correlation with its intervention in cold stress, suggesting its focal depletion in severe hypoxia determined by prolonged exposure to low ambient temperatures. – hard to read, I do not fully understand the meaning of this sentence. (same 541-544, try to make it more readable)

Response: We agreed with the reviewer and specific changes were made in the Conclusion section (highlighted in red in text of the manuscript).

Thank you again for taking the time to review and comment on our manuscript.

Reviewer 2 Report

Thank you for allowing me another opportunity to revise this manuscript. Many of the suggestions regarding structure and size of the initial manuscript have been included. Good work has been done. Still there is some redundancy, overlap and non relevant information included that, once improved, would make the manuscript more concise. There are many examples but some of them include:

Almost  all information in lines 177-201 can be found in table 2; there is no need to repeat this; Table 2 - death area is already addressed in Table 1; the same for lines 211-225 and table 3. A critical and detailed reflection is needed to define which information can be included in table 3 and which in the text of the manuscript. It is either - or, not in both

l. 257-259 "the following .... frequency" is self-explaining and can be removed

In general many sentence are too long and take 5 - 6 lines in the proof. Try to avoid that a sentence is longer than 3 lines. 

There are several spots where there is a reference to the literature without a reference number (f.e lines 100, 501  ) or when it reads as if there is a multitude of references but only reflected in one or no reference (f.e. line 343 "constantly noticed" - 1 reference; l. 445 "in previous studies" - 1 reference; l 459 "Numerous studies" - 2 references; l.493 "different studies" - 1 reference ) At the same time, when a reference is mentioned, one does not need to state this, because it is clear from the reference number that this information is taken from the literature(f.e. l 442; l 463;  

Some sentences in the results relate to issues relevant for the discussion, F.e. l. 239-242 ; l. 310 - 312 (and needs references); 

Details

The two-decimal figures of percentages can be reduced to one-decimal figures all over the manuscript, both in text and tables

It makes no sense to address frequencies that are different but not statistically significant different (f.e. l 248; l 262)

Please be consistent in use one word for "place of death", "location of death" , "death area"

Figures 1C, 2A, 2B may need an area to indicate where to look

Author Response

Response to Reviewer 2 Comments

We appreciate the positive general comments and the corrections required by Reviewer 2 to the manuscript and we have done all the necessary changes. Thank you again for your valuable comments.

Thank you for allowing me another opportunity to revise this manuscript. Many of the suggestions regarding structure and size of the initial manuscript have been included. Good work has been done. Still there is some redundancy, overlap and non relevant information included that, once improved, would make the manuscript more concise. There are many examples but some of them include:

  • Almost  all information in lines 177-201 can be found in table 2; there is no need to repeat this; Table 2 - death area is already addressed in Table 1; the same for lines 211-225 and table 3. A critical and detailed reflection is needed to define which information can be included in table 3 and which in the text of the manuscript. It is either - or, not in both

Response: We agreed with the reviewer and we have made the revision according to the comments. Table 2 has been revised in order to exclude the information related to the death area. The main text has been reformulated, deleted or shortened in 3.1. and 3.2 sections  (highlighted in red in the text of the manuscript). Thank you for this suggestion.

  • 257-259 "the following .... frequency" is self-explaining and can be removed

Response: The word the " following’’ has been removed. Thank you for pointing this out.

  • In general many sentence are too long and take 5 - 6 lines in the proof. Try to avoid that a sentence is longer than 3 lines.

Response: We agreed with the reviewer and specific changes were made main text. Please see the Result, Discussion and Conclusion sections (highlighted in red in the text of the manuscript). Thank you again for this suggestion.

  • There are several spots where there is a reference to the literature without a reference number (f.e lines 100, 501  ) or when it reads as if there is a multitude of references but only reflected in one or no reference (f.e. line 343 "constantly noticed" - 1 reference; l. 445 "in previous studies" - 1 reference; l 459 "Numerous studies" - 2 references; l.493 "different studies" - 1 reference ) At the same time, when a reference is mentioned, one does not need to state this, because it is clear from the reference number that this information is taken from the literature(f.e. l 442; l 463;  

Response: We agreed with the reviewer and we have made the revision according to the comments. 8 new references have been added in the main text and final list (highlighted in red in the manuscript). We appreciate these suggestions.

  • Some sentences in the results relate to issues relevant for the discussion, F.e. l. 239-242 ; l. 310 - 312 (and needs references); 

Response: We agree with the reviewer and specific changes have been made. Please see the Discussion section (highlighted in red in the manuscript).

Details

  • The two-decimal figures of percentages can be reduced to one-decimal figures all over the manuscript, both in text and tables

Response: We agreed with the reviewer and specific changes were made, where appropriate (in the main text, tables, and figures). Thank you for pointing this out.

  • It makes no sense to address frequencies that are different but not statistically significant different (f.e. l 248; l 262)

Response: We agreed with the reviewer and we have deleted the frequencies where no significant statistical data have been registered. We appreciate this suggestion.

  • Please be consistent in use one word for "place of death", "location of death" , "death area"

Response: We agreed with the reviewer and we have made the revision according to the comments. We appreciate this suggestion.

  • Figures 1C, 2A, 2B may need an area to indicate where to look

Response: Figures 1C, 2A, and 2B have been revised according to the reviewer's comments. Thank you for pointing this out.

Thank you again for taking the time to review and comment on our manuscript.
